# Matrix Metalloproteinases as Potential Biomarkers and Therapeutic Targets in Liver Diseases

**DOI:** 10.3390/cells9051212

**Published:** 2020-05-13

**Authors:** Eline Geervliet, Ruchi Bansal

**Affiliations:** Translational Liver Research, Department of Medical Cell BioPhysics, Technical Medical Centre, Faculty of Science and Technology, University of Twente, 7522 NB Enschede, The Netherlands; e.k.geervliet@utwente.nl

**Keywords:** matrix metalloproteinases, liver diseases, matrix remodeling, biomarkers, therapeutics

## Abstract

Chronic liver diseases, characterized by an excessive accumulation of extracellular matrix (ECM) resulting in scar tissue formation, are a growing health problem causing increasing morbidity and mortality worldwide. Currently, therapeutic options for tissue fibrosis are severely limited, and organ transplantation is the only treatment for the end-stage liver diseases. During liver damage, injured hepatocytes release proinflammatory factors resulting in the recruitment and activation of immune cells that activate quiescent hepatic stellate cells (HSCs). Upon activation, HSCs transdifferentiate into highly proliferative, migratory, contractile and ECM-producing myofibroblasts. The disrupted balance between ECM deposition and degradation leads to the formation of scar tissue referred to as fibrosis. This balance can be restored either by reducing ECM deposition (by inhibition of HSCs activation and proliferation) or enhancing ECM degradation (by increased expression of matrix metalloproteinases (MMPs)). MMPs play an important role in ECM remodeling and represent an interesting target for therapeutic drug discovery. In this review, we present the current knowledge about ECM remodeling and role of the different MMPs in liver diseases. MMP expression patterns in different stages of liver diseases have also been reviewed to determine their role as biomarkers. Finally, we highlight MMPs as promising therapeutic targets for the resolution of liver diseases.

## 1. Introduction

The liver is the largest organ in the human body and performs a variety of functions, including metabolic functions and detoxification. Liver functions are mainly carried out by the hepatocytes, the parenchymal liver cells that are abundantly present in the liver and are involved in metabolic processing of proteins, carbohydrates and lipids; detoxification of xenobiotic compounds; and secretion of essential molecules [1]. The nonparenchymal cells of the liver include cholangiocytes (epithelial cells), sinusoidal endothelial cells, hepatic stellate cells (HSCs), Kupffer cells (KCs) and other immune cells, including lymphocytes, that interact with each other and hepatocytes via paracrine factors and regulate liver functions. Increasing evidences have suggested that the interactions between parenchymal and nonparenchymal cells are involved in the pathogenesis of liver diseases [2]. One other essential, although noncellular, component of the liver microenvironment is the extracellular matrix (ECM), which is composed of collagen, fibronectin, laminin, proteoglycans and matricellular proteins [3]. The ECM forms a fibrous scaffold for cells and provides structural support, compressive strength and elasticity and functions as a reservoir for signaling molecules. Besides providing mechanical and biochemical properties, it is also involved in maintaining tissue homeostasis and regulating cell differentiation, proliferation, migration and adhesion [3]. The ECM interacts with growth factors and cytokines, thereby driving morphogenesis and cellular functions. The ECM therefore forms a complex dynamic microenvironment that undergoes continuous regulated remodeling during development, differentiation and wound healing processes to maintain homeostasis and prevent disease onset and progression [3,4].

Among the nonparenchymal cells, HSCs are the key cell type in the liver. Also known as vitamin A-storing cells, they are located in the space of Disse, i.e., the space between parenchymal cells and sinusoidal endothelial cells. HSCs maintain liver homeostasis by preservation of hepatocyte mass, vasoregulation, ECM remodeling and liver regeneration; to a lesser extent they regulate metabolism and detoxification [5]. These cells regulate the ECM composition by secreting ECM proteins, matrix metalloproteinases (MMPs) and tissue inhibitors of MMPs (TIMPs). In a healthy liver, the amount of physiological ECM proteins (production and degradation) are in equilibrium [6]. However, during liver injury, quiescent HSCs undergo phenotypic transformation into highly proliferative, promitogenic, profibrogenic, proinflammatory and contractile myofibroblast-like cells producing an excessive amount of ECM. Liver injury further progresses to liver fibrosis resulting from an imbalance between the production (HSC-induced) and degradation (by MMPs) of ECM. MMP–TIMP balance is also disturbed during liver injury, with increased TIMP expression by activated HSCs thus inhibiting MMPs, thereby reducing matrix degradation with increased ECM production by activated HSCs (or myofibroblasts) [5,7]. Since the identification of HSCs by Kupffer in the 19th century, and with increasing understanding of the versatile functions of HSCs, tremendous efforts have been made towards targeting of these cells for the resolution of liver diseases. Besides targeting HSCs and reducing ECM production and fibrosis, another interesting, however relatively poorly explored, approach is induction of ECM degradation via an increased expression of specific MMPs or liver-specific delivery of MMPs [8]. In this review, we present an overview of different MMPs, the role of MMPs and their expression patterns in different liver diseases. We further summarize different MMPs that have been explored as biomarkers and therapeutic targets in liver diseases.

## 2. Pathogenesis of Liver Diseases

Liver disease affects millions of people, accounting for about 1.2 million deaths per year, and therefore poses a significant health-related burden worldwide [9]. Liver diseases are mainly caused by hepatitis B/C viral (HBV/HCV) infections (viral hepatitis); excessive alcohol consumption (alcohol-associated liver diseases (ALD), alcoholic steatohepatitis (ASH) or alcoholic hepatitis (AH)); and metabolic disorders (non-alcoholic fatty liver disease (NAFLD) or non-alcoholic steatohepatitis (NASH)) [10]. Other uncommon causes include drug overdose (drug-induced liver injury (DILI)), auto-immune disorders (primary biliary cholangitis (PBC) or primary sclerosing cholangitis (PSC)) and genetic factors (hemochromatosis or Wilson’s disease) [11]. Among other liver diseases, ALD and NAFLD are the commonest etiologies of chronic liver disease worldwide. Both ALD and NAFLD display a similar spectrum of pathologies ranging from steatosis, steatohepatitis and fibrosis to cirrhosis and/or hepatocellular carcinoma (HCC or liver cancer) [12]. Liver cirrhosis is the common and final pathological pathway resulting from different etiologies.

Hepatocellular damage followed by liver inflammation and fibrosis culminates into liver cirrhosis and/or HCC. Chronic exposure to insults (e.g., hepatotoxins) forces the hepatocytes to undergo apoptosis and necrosis, secreting reactive oxygen species (ROS) and proinflammatory growth factors, cytokines and chemokines, including platelet-derived growth factor (PDGF), vascular endothelial growth factor (VEGF), connective tissue growth factor (CTGF) and transforming growth factor-β (TGF-β). Secretion of these proinflammatory factors induces recruitment and activation of resident and infiltrating immune cells, especially macrophages (resident KCs and circulating monocytes), leading to chronic liver inflammation. The immune cells secrete proinflammatory and profibrogenic factors that activate quiescent HSCs, resulting in an excessive accumulation of ECM and loss of liver architecture and function [13,14]. Activated HSCs are mostly responsible for the increased deposition of ECM by producing an excessive amounts of ECM components, mainly collagen-I (col-I) and -III (col-III), and TIMPs while limiting the secretion of MMPs [5,15]. Portal fibroblasts also contribute to an excessive ECM accumulation, mainly by secretion of collagen, especially in cholestatic fibrosis [15,16,17].

While acute liver injury—in most cases—can be reversed, chronic liver injury often results in irreversible and progressive cirrhosis [18]. Based on repeated biopsies in patients, mild to moderate fibrosis has been shown to be reversible and possibly dictated by collagen crosslinks. During cirrhosis, the disproportionate accumulation of collagen and extensive maturation of crosslinks results in the tissue stiffening, scarring and irreversibility [19]. Over past decades, hepatologists have made significant progress in disease understanding, monitoring and disease management. Notably, in the context of viral hepatitis, discovery and characterization of hepatitis viruses led to the development of preventive therapies including vaccines, antiviral and immunomodulatory therapies, with clinical treatments now being available for HBV/HCV-driven liver diseases [20,21]. However, two main issues remain unresolved, i.e., the issue of noninvasive, precise and early diagnosis (or disease staging) as well as the issue of effective treatment against liver diseases. Currently, the only available treatment for the end-stage liver diseases is liver transplantation; however, the feasibility of transplantation is limited due to a limited number of available donor organs and several risks and complications associated with liver transplantation, including transplant rejection, bleeding, infections and long-term immunosuppressants [22,23,24]. Other treatments that mainly focus on removing the underlying cause (e.g., a reduction of alcohol consumption in ALD or a healthy diet in NAFLD that can slow the progression or even reverse early stage fibrosis) are, however, insufficient for advanced fibrosis [10,25].

## 3. Matrix Metalloproteinases (MMPs) and Their Role in Liver Disease

MMPs belong to the large metzincin superfamily that includes four subfamilies: matrixins (MMPs); astracins (bone morphogenetic protein 1/tolloid-like protein 1, BMP1/TLL proteins or procollagen C-endopeptidases and meprins); bacterial serralysins; and adamalysins (disintegrin metalloproteinases (ADAMs), disintegrin and metalloproteinase with thrombospondin motif (ADAMTS) and snake venom proteins, reprolysins) [26,27]. MMPs or matrixins are calcium-dependent zinc-containing endo-proteinases that degrade the ECM components, regulate ECM integrity and composition and play an important role in ECM-mediated signaling [4]. Besides MMPs, ADAMs and ADAMTS, serine proteinases including plasmin and cathepsin G are specialized in degrading ECM protein components and therefore are also involved in ECM remodeling [28].

Besides ECM components, MMPs can also cleave cell surface molecules and pericellular nonmatrix proteins, thereby regulating cell behavior [29]. Moreover, MMPs are able to cleave a variety of other regulatory molecules, including serine protease inhibitors, cytokines and chemokines, and hence are involved in several developmental processes such as trophoblast implantation, embryogenesis, bone growth, wound healing and tissue regeneration [30]. Altogether, MMPs regulate essential cellular processes such as proliferation, differentiation, migration, adhesion and apoptosis [4]. To date, 25 MMPs have been discovered in vertebrates, including 24 different MMPs in humans. The first MMP, MMP-1, was discovered by Jerome Gross and Charles Lapiere in 1962 and was observed to be involved in collagen triple helix degradation during tadpole tail metamorphosis [31]. MMPs participate in several physiological processes such as bone remodeling, neuronal growth, innate and adaptive immune responses, inflammation and angiogenesis [32]. Due to their versatile functions, MMPs together with TIMPs have been implicated in many different diseases including inflammatory and fibrotic diseases, arthritis, cardiovascular disorders, cancer and metastases [33].

Human MMPs can be categorized into six different groups based on their substrate specificity and homology: (i) collagenases (MMP-1, -8, -13); (ii) stromelysins (MMP-3, -10, -11, -17); (iii) gelatinases (MMP-2, -9); (iv) matrilysins (MMP-7, -26); (v) membrane-type MMPs (MMP-14, -15, -16, -17, -24, -25); and (vi) others (MMP-12, -19, -20, -21, -22, -23 (-23A and -23B), -27, -28) [4].

Genomic analyses have revealed that there are 24 distinct genes encoding different MMPs. Based on the structural diversity, MMPs can be classified into different categories, as depicted in Figure 1. (a) Archetypal MMPs, comprised of collagenases (MMP-1, -8, -13), stromelysins (MMP-3, -10) and other MMPs (MMP-12, -19, -20, -22, -27). Archetypal MMPs contain unique structural domains: a highly-conserved hemopexin domain that ensures the substrate specificity and interaction with endogenous inhibitors, a catalytic domain containing two zinc ions (Zn^2+^) and at least one calcium ion (Ca^2+^), a hinge joining the catalytic and hemopexin domains, an amino-terminal pro-peptide to maintain enzyme latency and a signal peptide that directs secretion from the cells. (b) Gelatinases (MMP-2, -9) resemble the structure of archetypal MMPs and contain a fibronectin domain. (c) Matrilysins (MMP-7, -26) also resemble archetypal MMPs in structure but lack a hemopexin domain. (d) Secreted MMPs (MMP-11, -21, -28) also resemble archetypal MMPs in structure and contain a furin-like cleavable domain. (e) Membrane-type MMPs (MMP-14, MMP-15, MMP-16 and MMP-24) are localized at the cell surface and resemble archetypal MMPs. They contain a furin-like cleavable domain and transmembrane domains, i.e., C-terminal TM-1 and cytoplasmic tail. (f) Membrane-type MMPs, including MMP-17 and MMP-25, resemble archetypal MMPs and contain a C-terminal glycosylphosphatidylinositol (GPI) membrane anchor. (g) Other MMPs (MMP-23A and MMP-23B) also resemble archetypal MMPs, lack a hemopexin domain and contain N-terminal TM-II and cytoplasmic C-terminal immunoglobulin-like cell adhesion molecule (Ig-CAM) domains [34,35].

MMPs are mostly secreted in the form of pro-enzymes that are subsequently activated in the extracellular space. They are produced by different cell types in the body, including epithelial cells; fibroblasts; endothelial cells; and inflammatory cells including monocytes, macrophages and neutrophils, and have been implicated in different physiological and pathological processes [35]. In the liver, all hepatic cells, e.g., hepatocytes, HSCs, hepatic macrophages (including resident KCs and infiltrated monocyte-derived macrophages (MoMFs)) and infiltrated leukocytes, are able to produce MMPs; however, among these, HSCs are the major producers [36,37]. Different MMPs have been implicated in the initiation, progression and resolution of liver diseases [4,7,38]. Although the underlying mechanisms are largely unknown, MMPs are known to be involved at different stages of liver diseases from liver injury, inflammation, fibrosis, cirrhosis and hepatocarcinogenesis to disease resolution and liver regeneration (Figure 2). Due to the differential expression of different MMPs at different disease stages, MMPs have also been investigated as ‘direct’ (reflecting ECM turnover) and ‘indirect’ (molecules released into the blood that reflect abnormal hepatic function) biomarkers for accurate diagnosis and staging of liver fibrosis. Below, different MMPs have been described with their respective role as biomarkers and involvement in different liver diseases (summarized in Table 1).

MMP-1, also called collagenase-1, cleaves both ECM and non-ECM substrates such as collagen, gelatin, laminin, complement C1q, interleukin 1 beta (IL-1β) and tumor necrosis factor alpha (TNF-α), thereby playing an important role in fibrotic and inflammatory processes [111]. MMP-1 has high affinity with and is able to degrade col-I and -III, the most abundant proteins in the fibrotic liver. MMP-1 can also activate MMP-2 and MMP-9. MMP-1 is constitutively expressed in normal livers, mainly expressed by HSCs and possibly by inflammatory cells (e.g., mast cells, KCs, monocytes) in the liver [7,57]. MMP-1 serum levels were found to be inversely related to the disease severity in patients with chronic hepatitis C infection [39]. Based on the results, the combination of col-III/MMP-1 ratio (CMR), alpha-fetoprotein (AFP), aspartate aminotransferase (AST)/alanine aminotransferase (ALT) ratio and platelet count has been suggested for F2–F4 staging of fibrosis in patients with chronic hepatitis C infection [40]. In NASH patients, an increased expression of MMP-1 in monocytes, KCs and HSCs has been observed in early NASH, but not in advanced NASH, suggesting an inverse correlation between MMP-1 levels and fibrosis progression in NASH patients [41]. In patients with chronic hepatitis C infection and NASH, MMP-1 serum levels were closely associated with the early fibrosis (necroinflammation and fibroinflammation), while no correlation was observed at the advanced stages of fibrosis [41,112]. MMP-1 allelic polymorphism has also been implicated in the development of HCC [42] (Table 1).

MMP-1 has been used in different fibrosis models for the evaluation of the therapeutic efficacy of different therapies [25,43,113]. It has been reported that MMP-1 overexpression attenuates fibrosis by promoting col-I degradation, alters the ECM network and thereby the cell–ECM interaction, induces hepatocyte proliferation and thus liver regeneration, and promotes HSC apoptosis and hence reduced collagen production [113]. MMP-1 also plays a crucial role in ECM degradation during the recovery phase of experimental liver fibrosis [114,115] (Figure 2).

MMP-2, also called gelatinase A, is localized in the vascular regions and involved in maintaining liver vascular homeostasis [4]. MMP-2 is expressed by various liver cells, most abundantly by HSCs and KCs, and is one of the widely studied enzymes in liver fibrosis. MMP-2 is secreted as pro-enzyme and activated by MMP-1 and MT1-MMP (MMP-14) [44,97]. Unlike collagenases (MMP-1 and MMP-13) that bind, unwind and cleave triple helical collagen fibrils into tropocollagen (a basic subunit of collagen fibril), gelatinases (MMP-2 and MMP-9) bind and cleave denatured tropocollagen, suggesting the sequential and synergistic behavior of collagenases and gelatinases in collagen degradation [116].

Several studies evidently correlated the expression of MMP-2 to the progression of fibrosis, regardless of the etiology (with no detectable expression in normal livers), implicating its profibrogenic properties [44,45,46]. Increased MMP-2 expression has been associated with chronic hepatitis, liver fibrosis, alcoholic liver cirrhosis, ischemia and reperfusion injury (IRI), biliary atresia (BA) (progressive fibroinflammatory cholangiopathy of infancy and a leading indication for pediatric liver transplantation) and HCC (epithelial–mesenchymal transition (EMT) and metastasis) (Table 1) [44,45,46,47,48,117]. Furthermore, increased levels of inflammatory mediators such as interleukin-1 (IL-1) greatly enhance the biosynthesis and secretion of MMPs, including MMP-2, while activated MMPs degrade and negatively regulate IL-1β activity [118]. In patients with ALD, Prystupa et al. assessed MMP-2 (and MMP-8 and MMP-9) as a potential biomarker and found that MMP-2 was elevated during all the stages in alcoholic liver cirrhosis, while MMP-8 and MMP-9 were only increased in advanced liver cirrhosis (stage C) [48].

Using MMP-2^-/-^ and MMP-2^+/+^ mice, Radbill et al. investigated the fibrotic properties of MMP-2 and observed an increased fibrotic profile of MMP-2^-/-^ mice, indicating its antifibrotic properties, presumably via inhibition of the collagen-I expression by activated HSCs [49]. Consistently, the results of Onozuka et al. showed that MMP-2 deficiency in mice accelerated toxin- and cholestasis-induced fibrosis by activation of HSCs [50]. Lastly, Cursio and colleagues examined the expression profile of different MMPs, amongst which MMP-2 was implicated in the initiation, progression and resolution of IRI-induced fibrosis, which was further confirmed by Kato et al. [47,70] (Figure 2). Conclusively, MMP-2 mRNA expression was upregulated during all stages of fibrosis; however, only active MMP-2 was detected in the later stages, i.e., during the resolution of fibrosis, suggesting the role of MMP-2 in the resolution of hepatic IRI (and possibly other liver diseases).

MMP-3, also known as stromelysin-1, is secreted as pro-MMP-3 and gets activated by serine proteases. MMP-3 possesses the ability to cleave a wide range of substrates, including proteoglycans, fibronectin, laminin, gelatin and different types of collagens present in the ECM [53]. MMP-3 also plays a significant role in the activation of other MMPs (MMP-1 and MMP-9) [53] and seems to be involved during the early stages of fibrosis, presumably during the inflammation process [47]. This also explains why MMP-3 is not only expressed by HSCs but is also expressed by infiltrated macrophages during inflammation [57]. Increased expression of MMP-3 has been observed during liver inflammation, IRI-induced liver injury and HCC (and metastasis) (Table 1). However, the expression of MMP-3 by HSCs and macrophages was found to be significantly decreased during the resolution of fibrosis [47,53,57]. MMP-3 polymorphism (genetic variants) has been associated with poor HCC prognosis and primary sclerosing cholangitis (PSC), a cholestatic liver disease characterized by chronic inflammation and progressive inter- and intrahepatic biliary fibrosis [52,53,55]. Furthermore, in HCC, MMP-3 plays a pivotal role in the HCC invasiveness, EMT and metastasis [52,54,56], with higher expression levels observed in the tumor cells as well as adjacent to the blood vessels [51] (Figure 2).

MMP-7, also known as matrilysin-7, is mainly secreted by bile ductular epithelial cells, KCs and periportal hepatocytes and is a major MMP that is highly increased during BA-associated liver fibrosis. Positive correlation between MMP-7 and fibrosis stages was observed in BA patients [46]. In another recent study, intrahepatic MMP-7 expression was uniquely upregulated in BA patients after successful portoenterostomy, was localized in biliary epithelium and periportal hepatocytes and was correlated with MMP-7 serum levels and fibrosis stage [58]. These studies suggest MMP-7 as a potential postoperative prognostic marker for BA [119,120]. Since MMP-7 is solely and specifically involved in ECM remodeling during BA-associated fibrosis, a pharmacological therapeutic inhibiting MMP-7 might be an effective and promising therapy [46,58]. Moreover, among all MMPs, MMP-7 has the highest ability to degrade the highly cross-linked elastin found in elastic connective tissue such as blood vessels. The aptitude to degrade vascular basement membrane indicates the potential to facilitate hematogenous metastasis to the portal vein terminating in the liver (Figure 2). Zeng et al. investigated and confirmed the correlation of MMP-7 expression produced by tumor cells at different levels, including mRNA, protein and enzyme activity in colorectal carcinoma liver metastasis patients [59]. Furthermore, MMP-7 has shown to be expressed by liver progenitor cells (also known as oval cells) and regulates oval-cell-mediated regeneration, together with CD44, during liver injury (Table 1) [60].

MMP-8, also known as collagenase-2, mainly produced by neutrophils (and macrophages), is involved in the resolution of fibrosis by degrading collagen and plays an important role in controlling inflammatory responses by cleaving different cytokines and chemokines [121]. In TNF-induced lethal hepatitis, MMP-8-deficient mice showed impaired leukocyte influx and neutrophil-specific chemokine release implicating that, rather than influencing the hepatitis-induced hepatocyte necrosis as previously thought, MMP-8 plays a pivotal role in regulating acute inflammation (Figure 2) [65]. MMP-8 levels and activity (along with MMP-2 and MMP-9) have shown to be elevated in alcoholic patients with stage C liver cirrhosis [48]. Moreover, MMP-8 expression has been found to be associated with enhanced hepatocyte growth factor expression and hepatocyte proliferation [121]. Therapeutic effects of adenoviral-mediated MMP-8 overexpression have been widely investigated in different animal models and have been shown to induce fibrosis resolution [61,62,63,64]. Interestingly, Harty et al. showed that infiltrated neutrophils are responsible for MMP-8 expression during cholestatic liver injury repair (Table 1) [61]. MMP-8 has also been implicated in HCC by promoting tumor cell invasion, EMT and migration (Figure 2) [66].

MMP-9, also known as gelatinase-B, is secreted by a wide number of cell types, including neutrophils, macrophages and fibroblasts, and can degrade several ECM proteins, including collagen-IV, elastin and fibronectin. The interaction of fibronectin with its specific receptor, integrin alpha 4 beta 1 (α4β1), upregulates the expression and activity of MMP-9 during IRI. MMP-9 was found to be involved in specific ECM degradation and increases matrix permeability and chemotactic ECM fragments, thereby enhancing leukocyte infiltration and inflammation leading to impaired liver function during IRI [67]. MMP-9 inhibition therefore can be a promising approach for the treatment of IRI, as also reported by others, where MMP-9 deficiency or specific MMP-9 inhibitors showed reduction in leukocyte infiltration, hence inhibiting liver inflammation and damage [69,70]. In addition, inhibition of MMP-9 expression via inducible nitric oxide synthase (iNOS) deficiency (and reduced nitric oxide (NO) release) attenuated leukocyte infiltration, inflammation and liver damage [68]. Lastly, Wang et al. demonstrated that liver-selective MMP-9 inhibition prevents MMP-9-mediated VEGF cleavage that enhances recruitment and engraftment of bone-marrow-derived sinusoidal endothelial cell progenitor cells, thereby ameliorating IRI and accelerating liver regeneration [71]. Furthermore, MMP-9 inhibition has been shown to attenuate liver fibrogenesis, while KC-derived MMP-9 treatment promotes fibrosis resolution (Figure 2). MMP-9 has also shown to regulate the fate of HSCs via alpha v beta 3 (αvβ1) integrin, where MMP-9 induces HSC apoptosis [74]. Interestingly, MMP-9 activity has shown to be modulated by matrix stiffness, where increased fibrotic ECM downregulates MMP-9 expression, secretion and activity during fibrosis (and HCC) [122]. Furthermore, MMP-9 levels and activity were highly increased in alcoholic patients with advanced stage C cirrhosis [48].

Elevated MMP-9 expression was already associated with poor HCC prognosis in 1996; even though the mechanism had not been discovered, the correlation of MMP-9 expression and HCC capsule infiltration was stated [72]. Later studies were focused on gaining insights into the function of MMP-9. Sun et al. discovered that MMP-9 is secreted by fibroblasts and endothelial cells, has a high potential to degrade collagen-IV, a major structural component of basement membrane, and plays an imperative role in the formation of new capillary sprouts and neo-angiogenesis, eventually resulting in invasion and metastasis. They further found that iNOS-derived NO secreted by cancer cells modulates MMP-9 production, thereby contributing to tumor cell angiogenesis, invasion, EMT process and HCC metastasis (Figure 2) [73]. More studies have found the association of MMP-9 expression with poor HCC prognosis, as suggested by significant correlation with number of nodules, tumor size and differentiation, vascular and portal vein invasion as well as prediction of tumor recurrence and patient survival after surgical resection [75]. These studies therefore implicate MMP-9 as a suitable predictive marker for HCC prognosis. Besides HCC, MMP-9 expression was found to be highly increased during acute liver failure (ALF) and fulminant liver failure ((FLF) defined as acute and severe impairment of liver functions) and contributed to brain extravasation and edema due to the loss of blood–brain barrier (BBB) integrity [77]. Indeed, MMP-9 depletion or inhibition improved survival, liver functions and brain injuries at the early stage (unfortunately not at the late stage 3) [78] (Table 1).

MMP-10, also known as stromelysin-2, is secreted by both parenchymal and nonparenchymal liver cells, has a broad spectrum of substrates, including ECM components, and activates MMPs (MMP-1, -7, -8, -9 and -13). MMP-10 expression was reported to be markedly upregulated during acute liver injury, liver cirrhosis and HCC [4,51]. MMP-10 deficiency in mice revealed increased liver injury and impaired resolution during cholestatic liver injury and after partial hepatectomy [79]. Garcıa-Irigoyen and Latasa et al. demonstrated that MMP-10 contributes to HCC progression, EMT and metastasis via C-X-C chemokine receptor 4/stromal derived factor 1 (CXCR4/SDF1) axis (Figure 2 and Table 1) [80].

MMP-11, also known as stromelysin-3, has been associated with HCC (Figure 2). It has been shown that MMP-11 expression controls miR-125a-regulated proliferation, EMT and HCC metastasis [81]. Another study demonstrated that several MMP-11 gene variants are associated with and can reliably predict early-stage HCC and therefore can be used as a biomarker for HCC progression [82]. Interestingly, studies using MMP-11 transgenic (MMP11-Tg) mice and MMP11^-/-^ mice revealed that MMP-11 protects against diabesity (diabetes and obesity) and hepatic steatosis by controlling energy metabolism and increasing fat mobilization and metabolism [83]. In IRI, increased MMP-11 expression suggested that MMP-11 plays a distinct role after liver injury, perhaps during liver regeneration (Table 1) [47].

MMP-12, also called metalloelastase, is secreted by macrophages and is upregulated in schistosomiasis and cirrhosis [84]. In HCC, MMP-12 overexpression has been correlated with tumor size, AFP and poor overall survival, suggesting MMP-12 as an independent prognostic predictor for HCC (Figure 2) [85]. MMP-12 expression was found to be increased after 24 h of IRI and seems to play a critical role in macrophage-mediated ECM degradation and wound repair [39]. MMP-12 has also shown to degrade elastin during progressive fibrosis [86]. MMP-12 also inhibits ECM-degrading MMPs (MMP-2, -9 and -13), thereby aggravating IL-13-dependent fibrosis (Table 1) [84].

MMP-13, also known as collagenase-3, is the most well-studied MMP and the main collagen-degrading enzyme in rodents. Previously, MMP-13 was shown to play a pivotal role in acute liver injury [47,87], while later a study by Uchnami et al. demonstrated that loss of MMP-13 ameliorates cholestasis-induced liver inflammation and fibrogenesis [88]. In this study, authors suggested that the degradation of collagen alters the cell–matrix and cell–cell interactions, resulting in hepatocyte necrosis and apoptosis and promoting the release of chemotactic cytokines, thereby enhancing leukocyte infiltration and inflammation [88]. George et al. further investigated these mechanisms, revealing that the ECM degradation during early-stage fibrosis induces HSC proliferation and transdifferentiation into myofibroblast-like cells and that MMP-13-mediated CTGF cleavage induces acute inflammatory response [89]. Conversely, studies have reported that MMP-13, mainly produced by macrophages, is associated with the resolution of liver fibrosis [90,91]. This fibrolytic effect is attributed to the collagen, the most abundant ECM component in liver fibrosis [92]. However, another study stated that the overexpression of MMP-13 enhances recovery by inducing hepatic growth factor (HGF), MMP-2 and MMP-9 expression, which in turn are responsible for collagen degradation and hepatic repair [93]. MMP-13 levels were also increased in alcoholic patients with stage A, B and C cirrhosis and therefore suggested as a diagnostic marker for alcoholic liver cirrhosis [94]. In HCC, MMP-13 levels were found to be increased and were positively correlated with HCC progression and metastasis (Figure 2 and Table 1) [95,96].

MMP-14, also called MT1-MMP, contains a chain of hydrophobic amino acids that anchor the molecule in the membrane of macrophages and leukocytes, among other cells [123]. MMP-14 is associated with the degradation of several adhesion molecules, including fibronectin, and activates MMP-2 and MMP-13 [44,97]. MMP-14 enhances the recruitment of macrophages and leukocytes during IRI [98]. MMP-14 was found to be highly expressed in HCC tissue compared to nontumorous tissue and was positively correlated with invasion, metastasis and HCC recurrence [99]. Furthermore, MMP-14 assists metastasis by degrading ECM compounds during tumor invasion; however, possibly more importantly, MMP-14 enables cell survival after detachment that increases success of metastasis significantly (Figure 2 and Table 1) [100].

MMP-15, also known as MT2-MMP, can activate MMP-2 and MMP-13, thereby affecting ECM integrity [102]. MMP-15 was found to be downregulated during liver regeneration [101]. Indirect involvement of MMP-15 has been suggested during HCC tumor cell invasion (Table 1) [124].

MMP-16, also known as MT3-MMP, is localized on the surface of fibroblasts, can degrade ECM components such as collagen and can activate MMP-2. MMP-16 has shown to be expressed in hepatitis, cirrhosis and HCC. In HCC, MMP-16 promotes EMT, thereby supporting tumor cell invasion and metastasis; thus, it has been proposed as a prognostic marker in HCC (Table 1) [103,104].

MMP-19, also known as RASI-1, is widely expressed in the liver and has been associated with hepatic fibrosis. MMP-19^-/-^ mice showed reduced hepatic fibrosis via diminished ECM remodeling and accelerated liver regeneration regulated by TGF-β signaling pathway (Table 1) [105].

MMP-23 is an MMP that contains N-terminal transmembrane domain type II for anchoring the protein to the cell surface and C-terminal immunoglobulin-like cell adhesion molecule (Ig-CAM) domain (Figure 1) to mediate protein–protein and protein–lipid interactions. MMP-23 also contains a small toxin-like domain (TxD) that regulates intracellular trafficking of the potassium channel, thereby controlling the cellular functions. Hypermethylated MMP-23 has shown to be upregulated in the Mdr2^-/-^ mouse model of inflammation-mediated HCC and is proposed to be involved in long-term inflammation-driven hepatocarcinogenesis [106]. Furthermore, MMP-23b has shown to promote liver regeneration and hepatocyte proliferation mediated by TNF pathway, as investigated in the zebrafish model (Table 1) [107].

MMP-24, also called MT5-MMP, mediates cleavage of N-cadherin (CDH2) and thereby is involved in cell–cell interactions. PCR analysis of regenerating livers revealed an upregulation of MMP-24, suggesting the role of MMP-24 in liver regeneration (Table 1) [101].

MMP-25, also known as MT6-MMP, contains a glycosyl phosphatidyl inositol (GPI) anchor to attach to the plasma membrane (Figure 1) and has been implicated in tumor invasion via the nuclear factor kappa-light-chain-enhancer of activated B cells (NF-κB) signaling pathway (Table 1) [108].

MMP-28, also called epilysin, has shown to be upregulated during alcohol-mediated hepatocyte damage [8,109]. Upregulation of MMP-28 has been observed in HCC to induce migration and invasion and to be associated with poor prognosis, regulated by neurogenic locus notch homolog protein 3 (NOTCH-3) signaling pathway (Table 1) [110].

## 4. MMPs as Therapeutic Targets

MMPs play a pivotal role in ECM remodeling in normal physiology; however, dysregulation of MMP expression and/or activity has been found to directly or indirectly contribute to progression of liver diseases. Different MMPs are involved in the different stages, with their expression varying among acute liver injury, liver inflammation, fibrosis/cirrhosis and hepatocellular carcinoma, while some MMPs are also involved in the disease resolution (Figure 2). Preclinical animal studies using gene specific (specific MMP) knockout animal models have revealed the functions and the underlying mechanisms of specific MMPs in liver diseases. Furthermore, clinical studies have provided the insights into their expression levels in different liver pathologies. However, while most of the studies are focused on the understanding of MMPs in liver disease progression, only few studies have focused on the resolution or treatment of liver diseases. Here, we review the studies that have investigated MMPs as potential therapeutic targets for the resolution of liver diseases.

Several MMPs can degrade fibrillar collagens; among these, the most potent MMPs are collagenases, including MMP-1, MMP-8 and MMP-13. Delivery of MMP-1 for the resolution of liver diseases has been extensively studied and has shown to be a highly promising approach in different experimental animal models. In 2003, Iimuro et al. demonstrated that adenoviral-vector-mediated delivery of human pro-MMP-1 (Ad5MMP-1) attenuated established liver fibrosis in a long-term thioacetamide (TAA)-induced liver fibrosis rat model. In this study, the authors found that intravenous administration of Ad5MMP-1 resulted in intrahepatic pro-MMP1 expression and MMP-1 activity, suggesting that pro-MMP1 was successfully converted into the active form in vivo. Consequently, Ad5MMP-1 ameliorated fibrosis after two weeks and remained attenuated for an additional four weeks. Significantly, ECM degradation via MMP-1 delivery was accompanied by inhibition, disappearance and/or apoptosis of activated HSCs, subsequently resulting in reduced TIMP expression (thereby reducing ECM accumulation) and increased hepatocyte proliferation (favoring liver regeneration) [43]. This study suggested that the delivery of MMP-1 is a promising therapeutic approach to attenuate liver fibrosis. However, it is important to consider that the use of viral vectors can have immunogenic and toxic effects. When the vectors are used for a longer period, it is also possible that they may induce an unwanted overexpression of MMPs which may lead to adverse effects, for instance, degradation of normal physiological ECM and increased activation of other MMPs (e.g., MMP2 is one of the targets of MMP1 that induces ROS production and thereby might induce inflammation) [113].

In another study, Du et al. investigated a different MMP-1 delivery approach via transplantation of MMP-1-overexpressing bone-marrow-derived stem cells (BMSCs/MMP-1). In this study, isolated primary rat BMSCs were transfected with a recombinant adenoviral vector containing human MMP-1 gene, and these transfected cells were then transplanted in a carbon tetrachloride (CCl_4_)-induced rat model of liver fibrosis [25]. BMSCs/MMP-1 treatment resulted in decreased collagen levels and attenuated HSCs activation in fibrotic livers, subsequently resulting in amelioration of liver injury and fibrosis, indicating BMSCs/MMP-1 as a potential antifibrotic approach for the resolution of liver fibrosis. However, the use of BMSCs possesses some inherent challenges, e.g., invasive BMSC isolation procedure, poor survival rate/activity of BMSCs [125] and lack of optimized protocol for the delivery route or uncertainty about the number of injections [126]. Recently, Itaba et al. reported that orthotropic transplantation of IC-2 (a derivative of Wnt/β-catenin inhibitor) engineered BMSCs sheets inhibited chronic CCl_4_-induced liver fibrosis by inducing production of MMP-1 (and MMP-14 and thioredoxin) with subsequent suppression of HSCs activation [127]. In another study, Liu et al. explored MMP-1 induction by diethyldithiocarbamate (DDC) controlled by Akt and ERK/miR-222/ETS-1 pathways as a novel mechanism of MMP-1 regulation, suggesting that miR-222 inhibition (resulting in MMP-1 induction) is a potential approach for the treatment of liver fibrosis [128].

Besides MMP-1, MMP-8 delivery has been also explored using adenoviral vector. Garcia-Bañuelos et al. investigated the therapeutic efficacy of adenoviral vector mediated delivery of MMP-8 (AdMMP-8) in CCl_4_- and bile-duct ligation (BDL)- induced liver cirrhosis rat models. The authors found that in vivo administration of AdMMP-8 resulted in intrahepatic expression of pro-MMP-8 and its functional active form, consequently resulting in the reversal of fibrosis accompanied with an improvement in liver function tests and intrahepatic blood pressure in both animal models. Furthermore, authors found that TGF-β mRNA expression was decreased, while MMP-9 and HGF expression was highly increased. Overall, the results suggested AdMMP-8 as a promising antifibrotic therapy [64]. In another study, Liu et al. examined the effects of a fusion protein cMMP-8-1K1 containing MMP-8 and human hepatocyte growth factor mutant 1K1 in vitro and in vivo. cMMP-8-1K1 promoted hepatocyte proliferation and liver regeneration, ameliorated CCl_4_-induced liver fibrosis and protected liver function following 70% hepatectomy [62].

Apart from collagenases, gelatinases MMP-2 and -9 are also exploited as potential targets for the treatment of liver fibrosis. As mentioned previously, MMP-2 is involved in fibrosis progression as well as regression. Li et al. investigated HSC-specific delivery of MMP-2 siRNA using vitamin-A-coupled cationic liposomes. HSCs are known to express high levels of vitamin A receptors on their cell surface. Li et al. prepared vitamin-A-coupled liposomes to deliver MMP-2 siRNA (VitA-lip-MMP-2 siRNA) and demonstrated that VitA-lip-MMP-2 siRNA effectively delivered MMP-2 siRNA to the HSCs. VitA-lip-MMP-2 siRNA inhibited HSC activation and proliferation and subsequently decreased the deposition of collagen-I [129]. Although VitA-lip-MMP-2 siRNA showed promising results in vitro in HSC-T6 cells, no in vivo preclinical studies were performed to validate the in vitro findings. In another study, rosmarinic acid (RA) was examined and showed reversion of activated HSCs to quiescent phenotype; inhibition of MMP-2 activity; and suppression of reactive oxygen species (ROS), lipid peroxidation and oxidative stress [130]. The authors further explored the underlying mechanism of RA and showed that RA inhibits ROS-induced oxidative stress via nuclear factor erythroid 2-related factor 2 (NRF2) nuclear translocation; the RA-mediated effects on MMP-2 were found to be ROS-dependent, and RA counteracted HSCs activation via MMP-2 inhibition [130].

In another study, Roderfield et al. examined MMP-9 mutants for TIMP-1 scavenging properties. It is known that the imbalance in MMP/TIMP ratio regulates fibrosis development [38]. Interestingly, they showed that proteolytically inactive MMP-9 mutants as TIMP-1 antagonists attenuated CCl_4_-induced fibrosis, as confirmed by reduced portal and periportal deposition of collagen. MMP-9 mutants inhibited gene expression of fibrosis-related markers and suppressed HSC-to-myofibroblast transdifferentiation in vitro and in vivo. Furthermore, adenoviral-vector-mediated delivery of MMP-9 mutants led to increased HSCs apoptosis, as reported previously by vector-based delivery of MMPs. This study concluded that MMP-9 mutants are a novel therapeutic strategy for liver fibrosis by scavenging TIMPs [131].

Although MMPs have shown promising results in the preclinical models, none of the MMPs have been used as therapeutics in clinical trials for liver diseases. However, several clinical studies have explored MMPs as biomarkers and/or MMP genotype polymorphism as a risk factor in chronic liver diseases including HCC and colorectal liver metastasis. Before exploring MMPs as therapeutic targets, it is important to understand the role and mechanism of MMPs during the course of the disease. Both MMPs and MMPIs (MMP inhibitors) can be used as potential therapeutics. Although MMPs represent promising and interesting therapeutic targets, the following considerations should be kept in mind before designing MMPI or MMP delivery approaches: (i) MMPs are associated with normal physiologic processes, including ovulation, trophoblast invasion and embryonic development; (ii) ECM fragments resulting from MMP degradation are biologically active and therefore can also mediate secondary effects impacting the physiological and pathological processes; (iii) specificity and selectivity, poor pharmacokinetics, dose-limiting side effects/toxicity, instability and poor bioavailability should be considered with respect to MMPIs [132]; (iv) MMPs play a major role in immune processes, i.e., MMP-mediated cleavage activates and inhibits cytokines and chemokines; (v) increased MMP expression and activity have been clearly associated with cancer development and metastasis; (vi) MMP dysregulation affects ECM homeostasis, contributing to the aging process and neurodegenerative disorders [133]. Based on our current understanding of MMP pathophysiology and with considerable data available from clinical trials on cancer, new approaches targeting MMPs could contribute to the fight against chronic liver pathologies with unmet needs.

## 5. Conclusions

Liver injury due to excessive alcohol intake, unhealthy lifestyle or other risk factors that progresses to fibrosis and eventually cirrhosis and HCC affects millions of people worldwide and is the major cause of liver-related mortality and morbidity. Surprisingly, no therapeutically effective therapies are available, with liver transplantation representing the only treatment available for the end-stage liver failure. This review shines light on important mediators of liver diseases, the MMPs. MMPs play an important role in not only the initiation and progression, but also in the resolution of different liver diseases. Insight in the specific roles and expression patterns of different MMPs elucidates possibilities for new noninvasive biomarkers and therapeutic targets. Different MMPs and their inhibitors have been explored in preclinical studies for the treatment of liver diseases by degrading the most abundant fibrotic ECM protein, col-I. On the other hand, the serum levels of MMPs have been used as noninvasive biomarkers for staging liver diseases, including BA, ALD-induced fibrosis, HCC and metastasis. These examples demonstrate the importance of gaining knowledge about the possibilities of using MMPs for the diagnosis and treatment of liver diseases.

## Figures and Tables

**Figure 1 cells-09-01212-f001:**
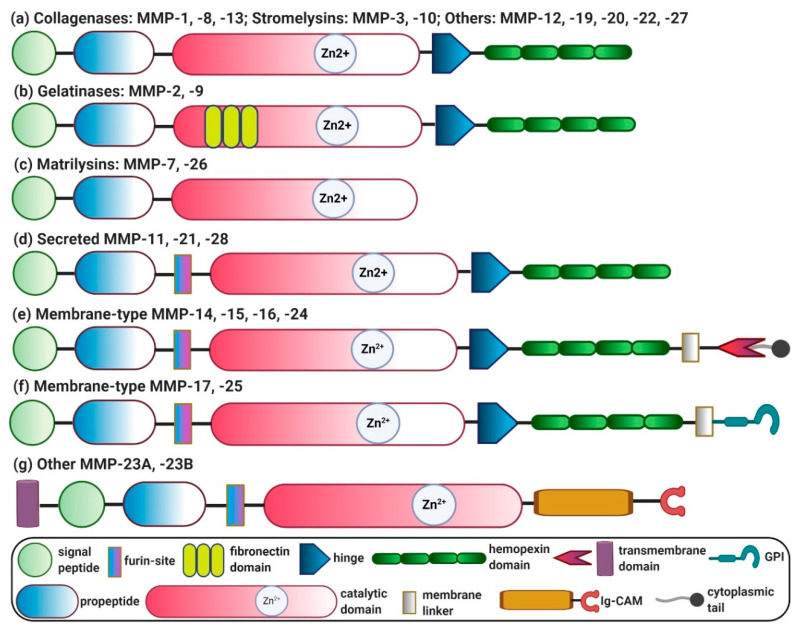
Domain structure diversity of human matrix metalloproteinases (MMPs), displayed as a schematic overview of different human MMPs categorized into groups based on their domain structure: (**a**) collagenases; (**b**) gelatinases; (**c**) matrilysins; (**d**) secreted MMPs; (**e**) membrane-type MMPs with transmembrane domains, C-terminal TM-1 and cytoplasmic tail; (**f**) membrane-type MMPs with C-terminal glycosylphosphatidylinositol (GPI) anchor; and (**g**) other MMPs with N-terminal transmembrane domain-II (TM-II) and cytoplasmic c-terminal immunoglobulin-like cell adhesion molecule (Ig-CAM). Zn^2+^, zinc ions.

**Figure 2 cells-09-01212-f002:**
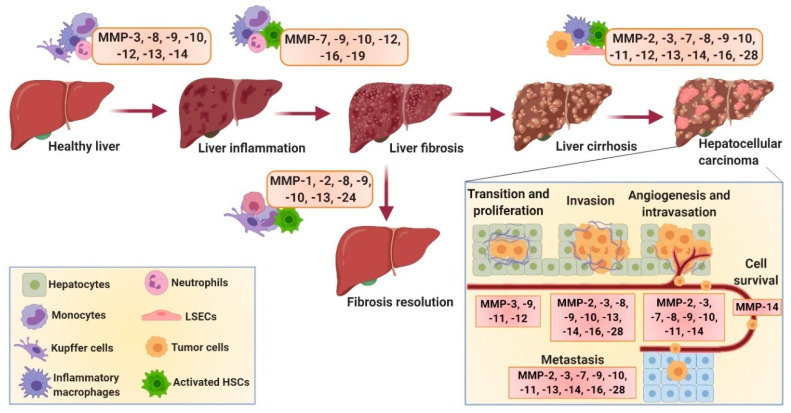
The role of different MMPs in the progression of liver diseases. Liver inflammation is induced by MMP-3, -8, -9, -10, -12, -13 and -14. These MMPs are involved in the degradation of normal ECM and the release of chemotactic cytokines that initiate macrophage and leukocyte infiltration and activation. MMP-7, -9, -10, -12, -16 and -19 are involved in fibrosis progression and ECM remodeling. When fibrosis is established, it can either be resolved directly by activation of ECM-degrading MMPs, such as MMP-1, -2, -8 and -13, or indirectly by MMP-10 and -24, or can lead to HCC regulated by MMP-2, -3, -7, -8, -9, -10, -11, -12, -13, -14, -16 and -28. HCC metastasis is a complex cascade consisting of endothelial-to-mesenchymal transition (EMT) and proliferation (MMP-3, -9, -11 and -12), invasion (MMP-2, -3, -8, -9, -10, -13, -14, -16 and -28), angiogenesis (MMP-9 and -10) and intravasation (MMP-2, -3, -7, -8, -9, -10, -11 and -14) into the bloodstream and extravasation (MMP-2, -3, -7, -9, -10, -11, -13, -14, -16 and -28) into other tissues.

**Table 1 cells-09-01212-t001:** Matrix metalloproteinases (MMPs) and their role in different hepatic diseases. ALF, acute liver failure; BA, biliary atresia; BBB, blood–brain barrier; BDL, bile duct ligation; CCl_4_, carbon tetrachloride; CD44, cluster of differentiation 44; FLF, fulminant hepatic failure; HCC, hepatocellular carcinoma; HSCs, hepatic stellate cells; IRI, ischemia reperfusion injury; MMP, matrix metalloproteinase; NASH, non-alcoholic steatohepatitis; PSC, primary sclerosing cholangitis; TNF, tumor necrosis factor.

MMP	Group	Role in Liver Diseases	Reference
MMP-1	Collagenase-1	Decreased expression is associated with disease severity in patients with chronic viral hepatitis and NASH; polymorphism is associated with HCC; induced expression attenuates liver fibrosis.	[25,39,40,41,42,43]
MMP-2	Gelatinase-A	Increased expression in chronic viral hepatitis, liver fibrosis, alcoholic liver cirrhosis (all stages), IRI, BA and HCC (and metastasis); depletion/inhibition exacerbates fibrosis in CCl_4_-, toxin-, cholestasis- and IRI-induced fibrosis.	[44,45,46,47,48,49,50]
MMP-3	Stromelysin-1	Increased expression during liver inflammation, IRI and HCC (and metastasis); polymorphism is associated with progressive liver disease, PSC and HCC; inhibition/depletion inhibits liver injury.	[47,51,52,53,54,55,56,57]
MMP-7	Matrilysin-7	Increased expression in BA-associated fibrosis, chronic hepatitis C, HCC and colorectal cancer liver metastasis; regulates oval-cell-mediated liver regeneration, together with CD44, during liver injury.	[46,58,59,60]
MMP-8	Collagenase-2	Increased expression in alcoholic liver cirrhosis (stage C); promotes HCC invasion and metastasis; involved in cholestatic liver injury repair; overexpression ameliorates CCl_4_-, TAA- and BDL-induced fibrosis; depletion protects against TNF-induced hepatitis.	[48,61,62,63,64,65,66]
MMP-9	Gelatinase-B	Increased expression in IRI, liver fibrogenesis, alcoholic liver cirrhosis (stage C), and HCC; induces HSCs apoptosis; promotes tumor invasion and metastasis; promotes fibrosis resolution; contributes to loss of BBB integrity and edema during ALF/FLF; depletion/inhibition ameliorates IRI, accelerates liver regeneration and improves early-stage brain injuries in ALF/FLF.	[48,67,68,69,70,71,72,73,74,75,76,77,78]
MMP-10	Stromelysin-2	Strongly expressed in acute liver injury, cirrhosis and HCC (and metastasis); deficiency impairs wound healing during cholestatic liver injury and after partial hepatectomy; contributes to HCC progression and metastasis.	[51,79,80]
MMP-11	Stromelysin-3	Genetic variants associated with HCC progression; regulates HCC proliferation and metastasis; protects against diabesity and hepatic steatosis; promotes liver regeneration in IRI.	[47,81,82,83]
MMP-12	Metalloelastase	Increased expression in schistosomiasis, cirrhosis, IRI and HCC; regulates elastin degradation in fibrosis; mediates wound healing in IRI; inhibits MMPs (MMP-2, -9 and -13), and aggravates fibrosis.	[47,84,85,86]
MMP-13	Collagenase-3	Increased in HCC and positively correlated with HCC progression and metastasis; promotes inflammation and fibrogenesis during acute liver injury; deficiency/depletion attenuates cholestasis-induced liver inflammation and fibrogenesis; macrophage-secreted MMP-13 mediates fibrosis resolution; overexpression enhances recovery and hepatic repair; overexpressed in alcoholic liver cirrhosis (all stages).	[47,87,88,89,90,91,92,93,94,95,96]
MMP-14	MT1-MMP	Highly expressed in HCC and positively correlated with invasion, metastasis and HCC recurrence; activates MMP-2 and -13; stimulates recruitment of macrophages and leukocytes in IRI.	[44,97,98,99,100]
MMP-15	MT2-MMP	Expression downregulated during liver regeneration; activates MMP-2 and MMP-13.	[101,102]
MMP-16	MT3-MMP	Expressed in hepatitis, cirrhosis and HCC (and metastasis).	[103,104]
MMP-19	RASI-1	Depletion reduces fibrosis and promotes regeneration.	[105]
MMP-23	Other	Involved in inflammation-driven hepatocarcinogenesis; promotes hepatocyte proliferation and regeneration.	[106,107]
MMP-24	MT5-MMP	Upregulation during liver regeneration.	[101]
MMP-25	MT6-MMP	Implicated in tumor invasion.	[108]
MMP-28	Secreted MMP (epilysin)	Upregulated during alcohol-mediated hepatocyte damage; upregulated and associated with migration, invasion and poor prognosis of HCC.	[109,110]

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
