# Peer review of "Matrix Metalloproteinases as Potential Biomarkers and Therapeutic Targets in Liver Diseases"

_cells, 2020, doi:10.3390/cells9051212_

Round 1

Reviewer 1 Report

Liver fibrosis remains a pharmacologically incurable disease. If allowed to progress to decompensated cirrhosis, the only available therapy is liver transplantation. However, due to the myriad of issues surrounding organ transplant (limited availability of donor organs, expense, life-long immunosuppression etc), it is clear additional therapies are needed. In this review article, Geervliet and Bansal propose MMPs as therapeutic targets in liver disease. The article has three main components. First, the role of MMPs in various liver diseases is reviewed. Second, the specific MMP expression patterns during liver disease progression are reviewed including their use as fibrosis biomarkers. Third, a discussion on MMPs as therapeutic targets is presented. The tables and figures are useful, informative, and (figures only) are artfully done. Overall, this is a comprehensive review and a nice update on the state of MMP knowledge in liver disease. I have a few comments the authors can consider as they revise their review.

Main comments.

  1. Section 2 would be benefit from reorganization. The first part is fine (extent of liver disease, etiologic agents), but then a clearer description of liver disease progression is warranted. Liver disease is a spectrum of disease states from fatty liver, to fibrosis, to cirrhosis, to HCC. Inflammation is a key component to each, but hepatitis per se can differ within disease etiologies (e.g. in alcohol-associated liver disease, acute hepatitis can be superimposed on any stage). As written, this disease spectrum is not clear. Moreover, the description of cirrhosis beginning at line 93 is confusing. I think the authors mean that cirrhosis consists of inflammation, fibrosis, and HCC and NOT that it contains three separate stages

  1. The review does much more than discuss MMPs as therapeutic targets. Therefore, the authors should revise their title to reflect its contents. (e.g. biomarkers is a large part of the discussion, as is general MMP biology and roles across the liver disease spectrum – certainly not just therapeutic targets.)

  1. The authors could consider discussing liver disease as ‘acute’ or ‘chronic’ and carefully distinguishing between the two states with respect to which MMPs would be good therapeutic targets. Stronger distinction between MMPs which should be inhibited and those which could be enhanced would be helpful and improve clarity.

  1. I recommend including the relationship between collagen and gelatin and thus the distinction between collagenase (MMP1 (humans, MMP13 mouse) and gelatinases (MMP2, 9) with respect to the sequential nature of collagen degradation.

  1. The authors should expand their list of caveats associated with targeting MMPs. For example, normal wound repair could be negatively impacted. How can you better target the therapies where they are needed? Have any MMPs been targeted in clinical trials? If so, mention. If not, the authors should mention that all available evidence or support for targeting MMPs comes from animal studies. Finally, some ECM fragments are very biologically active – could this be a concern if MMP-mediated ECM degradation is enhanced therapeutically?

  1. Figure 2 (which is beautiful), looks as thought it was stretched vertically. The aspect ratio should be maintained when trying to fit figures/schema within a defined space.

  1. Many references are missing. In several places, the message “Error! Reference source not found” is present instead of a reference number. In addition, significant details are missing (Lines 494-497). These issues must be addressed.

  1. I could not find the figures or tables mentioned in the text. Please add them to appropriate places in the text.

  1. As it is a shift in the field, I recommend that you define ALD as ‘alcohol-associated liver disease’.

Additional issues (not all grammar and usage mistakes are highlighted).

  1. While well-written in certain places, the manuscript would benefit from a revision of English grammar and usage. Avoid phrases such as “it has been shown that”, “it has been shown to be” etc. Strive for concise language.
  2. Line 13: apoptosis is not the only mode of hepatocyte cell death (which is pointed out in the body of the review). The abstract should not uniquely identify apoptosis in this way.
  3. Line 17: “…an emerge of scar tissue.” , seems incorrectly written. Please revise.
  4. Line 32: ‘xenophobic’ should be replaced with ‘xenobiotic’
  5. Line 33: Even though this review is focused on the liver, stellate cells should be called ‘hepatic stellate cells’
  6. Line 48: the word ‘habitat’ is odd as used here. Can say ‘located in the space of Disse’.
  7. Line 56: ‘myofibroblast-like’ (remove ‘s’)
  8. Line 63: I would remove ‘so called ECM producer part’. There is substantial literature supporting the statement that HSC are the main ECM producers when they are activated.
  9. Line 70: ‘diseases’ should be ‘disease’
  10. Line 78 – 79: note that fibrosis and even early cirrhosis is reversible. It seems that the crosslinking of collagen matrices is what dictates reversibility.
  11. Line 82: it might be worth mentioning the preventative strategies (vaccines) in the context of hepatitis infections (where relevant)
  12. Figure 1: I recommend moving the key down just a little bit, or placing the key within a box to separate it from the rest of the figure.
  13. Line 313: I would say “….parenchymal and nonparenchymal LIVER cells”
  14. Line 400: remove ‘evidences’ and replace with ‘evidence’

Author Response

Reviewer 1

Liver fibrosis remains a pharmacologically incurable disease. If allowed to progress to decompensated cirrhosis, the only available therapy is liver transplantation. However, due to the myriad of issues surrounding organ transplant (limited availability of donor organs, expense, life-long immunosuppression etc.), it is clear additional therapies are needed. In this review article, Geervliet and Bansal propose MMPs as therapeutic targets in liver disease. The article has three main components. First, the role of MMPs in various liver diseases is reviewed. Second, the specific MMP expression patterns during liver disease progression are reviewed including their use as fibrosis biomarkers. Third, a discussion on MMPs as therapeutic targets is presented. The tables and figures are useful, informative, and (figures only) are artfully done. Overall, this is a comprehensive review and a nice update on the state of MMP knowledge in liver disease. I have a few comments the authors can consider as they revise their review.

Response: We thank the reviewer for the kind words and for the critical comments that have been addressed in the revised version of the review. The changes made in the review have been highlighted in the revised version of the manuscript.

Main comments.

Section 2 would be benefit from reorganization. The first part is fine (extent of liver disease, etiologic agents), but then a clearer description of liver disease progression is warranted. Liver disease is a spectrum of disease states from fatty liver, to fibrosis, to cirrhosis, to HCC. Inflammation is a key component to each, but hepatitis per se can differ within disease etiologies (e.g. in alcohol-associated liver disease, acute hepatitis can be superimposed on any stage). As written, this disease spectrum is not clear. Moreover, the description of cirrhosis beginning at line 93 is confusing. I think the authors mean that cirrhosis consists of inflammation, fibrosis, and HCC and NOT that it contains three separate stages

Response: As suggested by the reviewer, section 2 has been improved as highlighted in the revised version of the review.

The review does much more than discuss MMPs as therapeutic targets. Therefore, the authors should revise their title to reflect its contents. (e.g. biomarkers is a large part of the discussion, as is general MMP biology and roles across the liver disease spectrum – certainly not just therapeutic targets.)

Response: We agree with the comment and therefore changed the title of the review to “Matrix metalloproteinases (MMPs) as potential biomarkers and therapeutic targets in liver diseases”

The authors could consider discussing liver disease as ‘acute’ or ‘chronic’ and carefully distinguishing between the two states with respect to which MMPs would be good therapeutic targets. Stronger distinction between MMPs which should be inhibited and those which could be enhanced would be helpful and improve clarity.

Response: We thank the reviewer for this critical comment. We made an effort to outline the involvement of different MMPs in different stages of liver diseases in Figure 2 and table 1. We sincerely hope that this will help to delineate the role of different MMPs in different disease stages.

I recommend including the relationship between collagen and gelatin and thus the distinction between collagenase (MMP1 (humans, MMP13 mouse) and gelatinases (MMP2, 9) with respect to the sequential nature of collagen degradation.

Response: We thank the reviewer for this critical comment. We have now made the following amendment in the revised version of the manuscript “Unlike collagenases (MMP-1 and MMP-13) that bind, unwind and cleave the triple helical collagen fibrils into tropocollagen (a basic subunit of collagen fibril), gelatinases (MMP-2 and MMP-9) bind and cleave denatured tropocollagens suggesting the sequential and synergistic behavior of collagenases and gelatinases in collagen degradation”.

The authors should expand their list of caveats associated with targeting MMPs. For example, normal wound repair could be negatively impacted. How can you better target the therapies where they are needed? Have any MMPs been targeted in clinical trials? If so, mention. If not, the authors should mention that all available evidence or support for targeting MMPs comes from animal studies. Finally, some ECM fragments are very biologically active – could this be a concern if MMP-mediated ECM degradation is enhanced therapeutically?

Response: We thank the reviewer for this comment and we sincerely think this is important to consider different caveats associated with targeting MMPs. The following has been appended in the revised version of the manuscript:

“Although MMPs have shown promising results in the preclinical models, none of the MMPs have been used as therapeutics in clinical trials for liver diseases. However, several clinical studies have explored MMPs as biomarkers and/or MMP genotypes polymorphism as a risk factor in chronic liver diseases including HCC and colorectal liver metastasis. Before exploring MMPs as therapeutic targets, it is important to understand the role and mechanism of MMPs during the course of the disease. Both MMPs and MMPIs (MMP inhibitors) can be used as potential therapeutics. Although MMPs represent as an promising and interesting therapeutic targets, the following considerations should be kept in mind before designing MMPIs or MMPs-delivery approaches: (i) MMPs are associated with normal physiologic processes including ovulation, trophoblast invasion and embryonic development; (ii) ECM fragments resulting from MMPs degradation are biologically active and therefore can also mediate secondary effects impacting the physiological and pathological processes; (iii) specificity and selectivity; poor pharmacokinetics, dose-limiting side effects/toxicity, instability and poor bioavailability should be considered with respect to MMPIs; (iv) MMPs play a major role in immune processes i.e. MMPs-mediated cleavage activates and inhibits cytokines and chemokines; (v) increased MMPs expression and activity have been clearly associated with cancer development and metastasis; (vi) MMPs dysregulation affects ECM homeostasis that contribute to the aging process and neurodegenerative disorders. Based on our current understanding of MMP pathophysiology and with considerable data available from clinical trials on cancer, new approaches targeting MMPs could contribute to the fight against chronic liver pathologies with unmet needs”.

Figure 2 (which is beautiful), looks as though it was stretched vertically. The aspect ratio should be maintained when trying to fit figures/schema within a defined space.

Response: As suggested by the reviewer, we have now made the recommended correction in the revised version of the manuscript.

Many references are missing. In several places, the message “Error! Reference source not found” is present instead of a reference number. In addition, significant details are missing (Lines 494-497). These issues must be addressed.

Response: The “Error! Reference source not found” originated from cross-references to figures and the table, which were lost due to the format change. These errors have been corrected in the revised version of the review.

I could not find the figures or tables mentioned in the text. Please add them to appropriate places in the text.

Response: The “Error! Reference source not found” originated from cross-references to figures and the table, which were lost due to the format change. These errors have been corrected in the revised version of the review.

As it is a shift in the field, I recommend that you define ALD as ‘alcohol-associated liver disease’.

Response: As suggested by the reviewer, we have now made the recommended correction in the revised version of the manuscript.

Additional issues (not all grammar and usage mistakes are highlighted). While well-written in certain places, the manuscript would benefit from a revision of English grammar and usage. Avoid phrases such as “it has been shown that”, “it has been shown to be” etc. Strive for concise language.

Response: As suggested by the reviewer, the review has been extensively revised for the possible mistakes.

Line 13: apoptosis is not the only mode of hepatocyte cell death (which is pointed out in the body of the review). The abstract should not uniquely identify apoptosis in this way.

Response: “During liver damage, injured hepatocytes release”

Line 17: “…an emerge of scar tissue.” , seems incorrectly written. Please revise.

Response: “leads to the formation of scar tissue referred to as fibrosis”

Line 32: ‘xenophobic’ should be replaced with ‘xenobiotic’

Response: ‘xenophobic’ is replaced with ‘xenobiotic’ in the revised version of the review.

Line 33: Even though this review is focused on the liver, stellate cells should be called ‘hepatic stellate cells’

Response: As suggested by the reviewer, we have now made the recommended correction in the revised version of the manuscript.

Line 48: the word ‘habitat’ is odd as used here. Can say ‘located in the space of Disse’.

Response: As suggested by the reviewer, we have now made the following correction in the revised version of the manuscript “Among the non-parenchymal cells, HSCs also known as vitamin A-storing cells, located in the space of Disse, the space between parenchymal cells and sinusoidal endothelial cells, are the key cell types in the liver”.

Line 56: ‘myofibroblast-like’ (remove ‘s’)

Response: As suggested by the reviewer, we have now made the recommended correction in the revised version of the manuscript.

Line 63: I would remove ‘so called ECM producer part’. There is substantial literature supporting the statement that HSC are the main ECM producers when they are activated.

Response: As suggested by the reviewer, we have now made the recommended correction in the revised version of the manuscript.

Line 70: ‘diseases’ should be ‘disease’

Response: As suggested by the reviewer, we have now made the recommended correction in the revised version of the manuscript.

Line 78 – 79: note that fibrosis and even early cirrhosis is reversible. It seems that the crosslinking of collagen matrices is what dictates reversibility.

Response: We agree with the reviewer and made the following amendment in the review:

“While acute liver injury - in most cases - can be reversed, chronic liver injury often results in irreversible and progressive cirrhosis. Based on repeated biopsies in patients, mild to moderate fibrosis has been shown to be reversible, and possibly dictated by collagen crosslinks. During cirrhosis, the disproportionate accumulation of collagen and extensive maturation of crosslinks results in the tissue stiffening, scarring and irreversibility”.

Line 82: it might be worth mentioning the preventative strategies (vaccines) in the context of hepatitis infections (where relevant)

Response: As suggested by the reviewer, we have now made the recommended correction in the revised version of the manuscript as follows:

“Notably in context to viral hepatitis, discovery and characterization of hepatitis viruses led to the development of preventive therapies including vaccines, anti-viral and immunomodulatory therapies with clinical treatments now being available for HBV/HCV-driven liver diseases [13], [14]”

Figure 1: I recommend moving the key down just a little bit, or placing the key within a box to separate it from the rest of the figure.

Response: As suggested by the reviewer, we have now made the recommended correction in the revised version of the manuscript.

Line 313: I would say “….parenchymal and nonparenchymal LIVER cells”

Response: As suggested by the reviewer, we have now made the recommended correction in the revised version of the manuscript.

Line 400: remove ‘evidences’ and replace with ‘evidence’

Response: As suggested by the reviewer, we have now made the recommended correction in the revised version of the manuscript.

Reviewer 2 Report

 The authors describe in their review, all MMPs and their roles in liver diseases. They cover multiple disease and discuss all MMPs and their functions in liver pathologies. However, there are a few points that need to be clarified:

1- MMPs are able to remodel the ECM but there are multiple other proteases that have been reported in the literature to also remodel the ECM. For example, Cathepsins, ADAMS, ADAMTS and others. I think it is important to mention alternative proteases that most likely play a key role as well. Although not all proteases are expressed in the liver, there are many of these proteases expressed in infiltrating immune cells and Kupffer cells. Therefore, it is likely that not only MMPs remodel the ECM but a multitude of proteases can regulate this. I understand that this is a focus on MMPs but 1-2 sentences could be added to acknowledge the roles of other metalloproteases.

2- line 114: there are 25 MMPs, not 26. MMP23 is a gene duplication and little is known about the differences so it is described as MMP23. The authors stated that there are 24 distinct genes encoding for MMPs so why are there 26? This should be rectified.

3- line 126, line 156, line 160, line 181, 188 etc.: many error citations should be corrected throughout the manuscript.

4- Table 1 and line 124: MMP23 is not considered a MT-MMP. There are 6 that have been described. Although it is likely that MMP23 is bound to the surface, it is not a Membrane-type MMP. This is misleading and described differently in all other MMP reviews.

5- line 192: MT-MMP (MMP14) should be MT1-MMP.

6- The authors overlooked that some MMPs are beneficial and should not be inhibited. This point needs to be discussed. Would a selective or broad-spectrum inhibitor be used? What are the best strategies to inhibit only the detrimental MMPs? If some MMPs are beneficial and others are detrimental, how can MMPS be therapeutic targets? The authors mention a viral delivery of MMPs but there is no known example of viral delivery of MMPs and is likely not be able to work in humans (well, maybe in the future but not now). And what MMP would be selected for viral delivery? more than 1 MMP enhance liver pathologies. This needs to be addressed.

7- MMPs cleave multiple other substrates than the ECM. I think it should be mentioned as it is short-sighted that MMPs are ECM remodeller. They have many other functions.

Author Response

Reviewer 2:

The authors describe in their review, all MMPs and their roles in liver diseases. They cover multiple disease and discuss all MMPs and their functions in liver pathologies.

Response: We thank the reviewer for the kind words and for the critical comments that have been addressed in the revised version of the review.

However, there are a few points that need to be clarified:

1- MMPs are able to remodel the ECM but there are multiple other proteases that have been reported in the literature to also remodel the ECM. For example, Cathepsins, ADAMS, ADAMTS and others. I think it is important to mention alternative proteases that most likely play a key role as well. Although not all proteases are expressed in the liver, there are many of these proteases expressed in infiltrating immune cells and Kupffer cells. Therefore, it is likely that not only MMPs remodel the ECM but a multitude of proteases can regulate this. I understand that this is a focus on MMPs but 1-2 sentences could be added to acknowledge the roles of other metalloproteases.

Response: We thank the reviewer for this comment. As suggested by the reviewer, we have now made the recommended correction in the revised version of the manuscript. We have now made the following amendment:

MMPs belongs to the large metzincin superfamily that includes four subfamilies: matrixins (MMPs), astracins (Bone Morphogenetic Protein 1/Tolloid-like protein 1, BMP1/TLL proteins or procollagen C-endopeptidases and meprins), bacterial serralysins, and adamalysins (disintegrin metalloproteinases, ADAMs; disintegrin and metalloproteinase with thrombospondin motif, ADAMTS and snake venom proteins, reprolysins) [26], [27]. MMPs or matrixins are calcium-dependent zinc-containing endo-proteinases that degrade the ECM components, regulate ECM integrity and composition, and play an important role in ECM-mediated signaling [4]. Besides MMPs, ADAMs, ADAMTS, serine proteinases including plasmin and cathepsin G are specialized in degrading ECM protein components therefore are also involved in ECM remodeling [28].

2- line 114: there are 25 MMPs, not 26. MMP23 is a gene duplication and little is known about the differences so it is described as MMP23. The authors stated that there are 24 distinct genes encoding for MMPs so why are there 26? This should be rectified.

Response: As suggested by the reviewer, we have now made the recommended correction in the revised version of the manuscript.

3- line 126, line 156, line 160, line 181, 188 etc.: many error citations should be corrected throughout the manuscript.

Response: The “Error! Reference source not found” originated from cross-references to figures and the table, which were lost due to the format change. These errors have been corrected in the revised version of the review.

4- Table 1 and line 124: MMP23 is not considered a MT-MMP. There are 6 that have been described. Although it is likely that MMP23 is bound to the surface, it is not a Membrane-type MMP. This is misleading and described differently in all other MMP reviews.

Response: As suggested by the reviewer, we have now made the recommended correction in the revised version of the manuscript.

5- line 192: MT-MMP (MMP14) should be MT1-MMP.

Response: As suggested by the reviewer, we have now made the recommended correction in the revised version of the manuscript.

6- The authors overlooked that some MMPs are beneficial and should not be inhibited. This point needs to be discussed. Would a selective or broad-spectrum inhibitor be used? What are the best strategies to inhibit only the detrimental MMPs? If some MMPs are beneficial and others are detrimental, how can MMPS be therapeutic targets? The authors mention a viral delivery of MMPs but there is no known example of viral delivery of MMPs and is likely not be able to work in humans (well, maybe in the future but not now). And what MMP would be selected for viral delivery? more than 1 MMP enhance liver pathologies. This needs to be addressed.

Response: We agree with the reviewer comment and think it is indeed important to highlight and provide an outlook about MMPs as potential diagnostic and therapeutic targets. Throughout the review, we have highlighted which MMPs have been targeted and which MMPs have been used as biomarkers as also mentioned in figure 2 and table 1. We have further added the following providing our outlook about current and future possible clinical use of MMPs

“Although MMPs have shown promising results in the preclinical models, none of the MMPs have been used as therapeutics in clinical trials for liver diseases. However, several clinical studies have explored MMPs as biomarkers and/or MMP genotypes polymorphism as a risk factor in chronic liver diseases including HCC and colorectal liver metastasis. Before exploring MMPs as therapeutic targets, it is important to understand the role and mechanism of MMPs during the course of the disease. Both MMPs and MMPIs (MMP inhibitors) can be used as potential therapeutics. Although MMPs represent as an promising and interesting therapeutic targets, the following considerations should be kept in mind before designing MMPIs or MMPs-delivery approaches: (i) MMPs are associated with normal physiologic processes including ovulation, trophoblast invasion and embryonic development; (ii) ECM fragments resulting from MMPs degradation are biologically active and therefore can also mediate secondary effects impacting the physiological and pathological processes; (iii) specificity and selectivity; poor pharmacokinetics, dose-limiting side effects/toxicity, instability and poor bioavailability should be considered with respect to MMPIs; (iv) MMPs play a major role in immune processes i.e. MMPs-mediated cleavage activates and inhibits cytokines and chemokines; (v) increased MMPs expression and activity have been clearly associated with cancer development and metastasis; (vi) MMPs dysregulation affects ECM homeostasis that contribute to the aging process and neurodegenerative disorders. Based on our current understanding of MMP pathophysiology and with considerable data available from clinical trials on cancer, new approaches targeting MMPs could contribute to the fight against chronic liver pathologies with unmet needs”.

7- MMPs cleave multiple other substrates than the ECM. I think it should be mentioned as it is short-sighted that MMPs are ECM remodeler. They have many other functions.

Response: We completely agree with this comment and therefore added the following in the revised version of the manuscript:

“Beside ECM components, MMPs can also cleave cell surface molecules and pericellular non-matrix proteins thereby regulating cell behavior [29]. Moreover, MMPs are able to cleave a variety of other regulatory molecules including serine protease inhibitors, cytokines and chemokines, hence are involved in several developmental processes such as trophoblast implantation, embryogenesis, bone growth, wound healing and tissue regeneration [30]. Altogether, MMPs regulate essential cellular processes such as proliferation, differentiation, migration, adhesion, and apoptosis [4]. To date, 25 MMPs have been discovered in vertebrates including 24 different MMPs in humans”.

Reviewer 3 Report

This review article entitled “Matrix metalloproteinases (MMPs) as therapeutic 2 targets in liver diseases” by Geervliet and Bansal explains that liver fibrosis (excessive ECM deposition) is caused by the activated hepatic stellate cells in response to various insults to the liver. It describes dysregulation of expression levels of different MMPs in different stages of liver fibrosis and cancer. Unfortunately, the version of this review submitted to the journal appears to be a preliminary draft as you can find a note “Error! Reference source no found” in numerous positions throughout the manuscript. While the topic of the article is clinically relevant and important, the manuscript requires major rewriting and refinement before considered for publication.

Major points

The important aspect of the therapeutic effects of MMPs is the ability to induce apoptosis/attenuation of activated HSCs. This is possibly due to lack of survival signaling provided by the ECM. Curing fibrosis is by degrading ECM is one thing, but underlying cellular mechanism to reverse inflammatory response should be discussed. The article should focus more of the pivotal signaling pathways manipulated by MMPs that help recovery from liver diseases.

In addition to the long list of individual MMP level that goes up and down by the different stages of liver diseases, a figure summarizes interactions between MMPs (such as MMP2&9 are activated by MMP1), and what has been successful as therapy should be included to help the audience to understand/overview the main point of the article.

As for diagnostic use of levels of MMPs for different stages of liver disease, it is hard to see how promising the idea is: First of all, many of them are overlapping in different stages. Secondly, the entire liver might not show same progression of the disease as liver is a huge organ. Any advantages over conventional measurement of liver function should be listed.

Minor points

Line127 should be correlated with Fig.1. There is no mention of figures or table in the main text. Fig. 1 should have labeling (a)-(g) as indicated in the main text.

Line169-Line170 requires references.

Line 199 should mention and cite references that inflammatory signaling such as IL1 induces MMP2 expression.

In Line427, it appears to be the problem is not simply the over-degradation of ECM but also causing the source of inflammation as MMP2, one of targets of MMP1 induces ROS.

In Line225, color coding in this figure is unclear. Why Kupffer cells in Fibrosis resolution stage are blue?

Author Response

Reviewer 3

This review article entitled “Matrix metalloproteinases (MMPs) as therapeutic targets in liver diseases” by Geervliet and Bansal explains that liver fibrosis (excessive ECM deposition) is caused by the activated hepatic stellate cells in response to various insults to the liver. It describes dysregulation of expression levels of different MMPs in different stages of liver fibrosis and cancer. Unfortunately, the version of this review submitted to the journal appears to be a preliminary draft as you can find a note “Error! Reference source no found” in numerous positions throughout the manuscript. While the topic of the article is clinically relevant and important, the manuscript requires major rewriting and refinement before considered for publication.

Response: We thank the reviewer for the kind words and for the critical comments that have been addressed in the revised version of the review.

The “Error! Reference source not found” originated from cross-references to figures and the table, which were lost due to the format change. These errors have been corrected in the revised version of the review.

Major points

The important aspect of the therapeutic effects of MMPs is the ability to induce apoptosis/attenuation of activated HSCs. This is possibly due to lack of survival signaling provided by the ECM. Curing fibrosis is by degrading ECM is one thing, but underlying cellular mechanism to reverse inflammatory response should be discussed. The article should focus more of the pivotal signaling pathways manipulated by MMPs that help recovery from liver diseases.

Response: We completely agree with this comment and therefore added the following in the revised version of the manuscript:

“Besides ECM components, MMPs can also cleave cell surface molecules and pericellular non-matrix proteins thereby regulating cell behavior [29]. Moreover, MMPs are able to cleave a variety of other regulatory molecules including serine protease inhibitors, cytokines and chemokines, hence are involved in several developmental processes such as trophoblast implantation, embryogenesis, bone growth, wound healing and tissue regeneration [30]. Altogether, MMPs regulate essential cellular processes such as proliferation, differentiation, migration, adhesion, and apoptosis [4]. To date, 25 MMPs have been discovered in vertebrates including 24 different MMPs in humans”.

In addition to the long list of individual MMP level that goes up and down by the different stages of liver diseases, a figure summarizes interactions between MMPs (such as MMP2&9 are activated by MMP1), and what has been successful as therapy should be included to help the audience to understand/overview the main point of the article.

Response: We understand the reviewer’s point-of-view and looking at the complexity, it is difficult to conclude this complexity with a figure. we have however provided this overview in Figure 2. We sincerely hope that figure 2 with a table is sufficient to demonstrate the involvement of different MMPs in different stages of liver diseases.

As for diagnostic use of levels of MMPs for different stages of liver disease, it is hard to see how promising the idea is: First of all, many of them are overlapping in different stages. Secondly, the entire liver might not show same progression of the disease as liver is a huge organ. Any advantages over conventional measurement of liver function should be listed.

Response: We understand the critical comment by the reviewer. Although we completely agree with the comment, we also think that an additional biomarker would be useful for the assessment of liver diseases. Considering this comment we have included this information in the revised version providing our overview on use of MMPs as diagnostic and therapeutic targets:

Although MMPs have shown promising results in the preclinical models, none of the MMPs have been used as therapeutics in clinical trials for liver diseases. However, several clinical studies have explored MMPs as biomarkers and/or MMP genotypes polymorphism as a risk factor in chronic liver diseases including HCC and colorectal liver metastasis. Before exploring MMPs as therapeutic targets, it is important to understand the role and mechanism of MMPs during the course of the disease. Both MMPs and MMPIs (MMP inhibitors) can be used as potential therapeutics. Although MMPs represent as an promising and interesting therapeutic targets, the following considerations should be kept in mind before designing MMPIs or MMPs-delivery approaches: (i) MMPs are associated with normal physiologic processes including ovulation, trophoblast invasion and embryonic development; (ii) ECM fragments resulting from MMPs degradation are biologically active and therefore can also mediate secondary effects impacting the physiological and pathological processes; (iii) specificity and selectivity; poor pharmacokinetics, dose-limiting side effects/toxicity, instability and poor bioavailability should be considered with respect to MMPIs [132]; (iv) MMPs play a major role in immune processes i.e. MMPs-mediated cleavage activates and inhibits cytokines and chemokines; (v) increased MMPs expression and activity have been clearly associated with cancer development and metastasis; (vi) MMPs dysregulation affects ECM homeostasis that contribute to the aging process and neurodegenerative disorders [133]. Based on our current understanding of MMP pathophysiology and with considerable data available from clinical trials on cancer, new approaches targeting MMPs could contribute to the fight against chronic liver pathologies with unmet needs.

Minor points

Line127 should be correlated with Fig.1. There is no mention of figures or table in the main text. Fig. 1 should have labeling (a)-(g) as indicated in the main text.

Response: The “Error! Reference source not found” originated from cross-references to figures and the table, which were lost due to the format change. These errors have been corrected in the revised version of the review.

Line169-Line170 requires references.

Response: As suggested by the reviewer, we have now made the recommended correction in the revised version of the manuscript.

Line 199 should mention and cite references that inflammatory signaling such as IL1 induces MMP2 expression.

Response: As suggested by the reviewer, the following correction has been made in the revised version of the manuscript:

“Furthermore, increased levels of inflammatory mediators such as interleukin-1 (IL-1) greatly enhances the biosynthesis and secretion of MMPs including MMP-2, while activated MMPs degrade and negatively regulate IL-1β activity”.

In Line 427, it appears to be the problem is not simply the over-degradation of ECM but also causing the source of inflammation as MMP2, one of targets of MMP1 induces ROS.

Response: We thank the reviewer for this comment. As suggested by the reviewer, the following has been added in the revised version of the manuscript:

“Also, when the vectors are used for a longer period, it is possible that they may induce an unwanted overexpression of MMPs which may lead to adverse effects, for instance, degradation of normal physiological ECM, increased activation of other MMPs e.g. MMP2 is one of the targets of MMP1 that induces ROS production thereby might induce inflammation”.

In Line225, color coding in this figure is unclear. Why Kupffer cells in Fibrosis resolution stage are blue?

Response: As suggested by the reviewer, we have now made the recommended correction in the revised version of the manuscript.

Round 2

Reviewer 2 Report

All comments were properly addressed.

Reviewer 3 Report

Revision seems adequately made, and my concerns are well addressed. Improvements are recognizable. There are no further comments to be made.